# Novel Nutraceutical Compounds in Alzheimer Prevention

**DOI:** 10.3390/biom12020249

**Published:** 2022-02-03

**Authors:** Ricardo Benjamin Maccioni, Camila Calfío, Andrea González, Valentina Lüttges

**Affiliations:** 1International Center for Biomedicine ICC, Vitacura 3568, Santiago 7630000, Chile; cdp.calfio@gmail.com (C.C.); azgonzal@uc.cl (A.G.); vluttges@uc.cl (V.L.); 2Laboratory of Neuroscience and Functional Medicine, Faculty of Sciences, University of Chile, Santiago 7630000, Chile

**Keywords:** AD prevention, interventional approaches toward Alzheimer´s treatment, nutraceutical compounds, mechanisms of action in AD therapy, bioactive molecules derived from plants in AD, vitamins and cofactors actions in AD

## Abstract

Alzheimer’s disease (AD) incidence is increasing worldwide at an alarming rate. Considering this increase, prevention efforts, stemming from scientific research, health education, and public policies, are critical. Clinical studies evidenced that healthy lifestyles along with natural multitarget and disease-modifying agents have a preventative impact on AD or mitigate symptoms in diagnosed patients. The pathological alterations of AD start 30 years before symptoms, and it is essential to develop the capacity to detect those changes. In this regard, molecular biomarkers that detect early pathological manifestations are helpful. Based on markers data, early preventive interventions could reduce more than 40% of AD cases. Protective actions include exercise, shown to induce neurogenesis, cognitive stimulation, intellectual-social activity, and nutrition among others. Mediterranean diet, preprobiotics, and nutraceuticals containing bioactive molecules with antioxidant and anti-inflammatory properties are relevant. Antiprotein aggregation molecules whose mechanisms were described are important. Anti-inflammatory agents with anti-aggregation properties that help to control cognitive impairment, include quercetin, biocurcumin, rosemarinic acid, and Andean *shilajit*. Anthocyanidins, e.g., delphinidin, malvidin, and natural flavonoids, are also included. Quercetin and hydroxy-tyrosol are antiaging molecules and could have anti-AD properties. We emphasize the relevance of nutraceuticals as a main actor in the prevention and/or control of dementia and particularly AD.

## 1. Introduction

Alzheimer’s Disease (AD) is a progressive neurodegenerative disease characterized by cognitive deterioration, mood alterations, and neuropsychiatric disorders. There are more than 52 million people worldwide affected by AD (Alzheimer Report WHO), while most of pharmacological agents have only some palliative actions [1]. Besides the deteriorating effects of AD on human health and the quality of life of elderly persons, there is also a tremendous economic impact associated with the disease. Economical cost of AD in the world reaches one billion dollars a year [2]. AD is not only a medical issue and a puzzle for society but also linked with public policies in the search for quality of life of patients and protection of caregivers for AD patients. In this context, we are promoting integrative action from basic and translational research to development of innovative technologies and actions in favor of caregivers.

Neuroinflammation is one of the major causes of Alzheimer’s disease. The mechanisms on how the inflammatory process occurs in the human brain starts with the so named “damage signals”, which interfere with the cross-talks neuron-glia. Consequently, activated microglia produce NFkB, leading the synthesis of proinflammatory mediators that finally signal on neuronal receptors, with reactivation of proteins kinases responsible for tau hyperphosphorylation. In a search of nutraceutical bioactive principles, we can find compounds with tau antiaggregant activity, as well as compounds with antioxidative and anti-inflammatory activities [3,4,5].

In this context, and considering the explosive increase in AD incidence, the path to AD prevention appears as a most promising avenue to control the spread of this disease [6]. Healthy lifestyles, along with several nonpharmacological actions, were demonstrated through clinical studies to prevent manifestations of the disease and even mitigate the symptoms of diagnosed AD patients [7]. These actions include cognitive and sensory stimulation, mindfulness, practice of Chinese medicine, the Ayurveda, and especially nutrition. In the latter set of actions, the use of nutraceuticals appears to be of enormous relevance and one of most effective preventive actions [6]. On the other hand, these approaches need to be accompanied by using early detection tools including molecular biomarkers. Early detection of cognitive impairment in asymptomatic patients constitutes a warning alarm to promote the use of nutraceuticals [8,9,10,11].

Functional foods are those that are considered beneficial for health and that go beyond simple nutrition: some in their natural form, such as fish or vegetables; others are preparations such as preprobiotics that are important in protecting the organism against chronic diseases and/or pathological disorders [12]. Many bioactive compounds are present in food but at low concentrations, such as flavonoid and anthocyanins in fruits and vegetables, but if used in a concentrate preparation these can be nutraceutical products that strongly contribute to the integral health of individuals [5,13,14]. Smart “drugs” quickly boost cognition. For those seeking a natural approach, four plant extracts improve brain processing speed, memory, learning, and mental concentration: blueberries, rosemary, curcuma, and garlic.

A nutraceutical is a biopharmaceutical product of natural ingredients that exhibits reliably beneficial actions in human health. This includes medicinal products made with natural ingredients. New nutritional trends and the need to meet social and health demands drove the increasing demand of functional and nutraceutical foods that, in addition to their general nutritional functions, have properties for maintaining health and longevity. One of the main challenges facing this revolution of nutraceutical foods is the absence of a single and universal definition, as well as a legal regulation of them.

Vitamins and derivatives have also neuroprotective actions helping in AD prevention. Neuroprotective actions against neuronal death were evidenced for vitamins B6, 9, and B12 [15]. Besides, the derivative of B3 Nicotinamide riboside, also called niacin, is effective. Like other forms of vitamin B3, nicotinamide riboside is converted by the cells into nicotinamide adenine dinucleotide (NAD+), a coenzyme, or helper molecule. There is evidence that NAD+ slows down the progress of AD in patients at an initial stage of the disease [16].

## 2. Methods

A systematic search for relevant bioactive compounds or nutraceuticals linked to prevention of AD was performed according to the guidelines and items required for Systematic Reviews. The following electronic databases were used to identify pertinent publications: Web of Science, PubMed^®^, Springer and Google Scholar. The literature search was conducted within the period September–November 2021. Combinations of the search terms “bioactive compounds”, “nutraceuticals” and/or “Alzheimer’s disease” were used. In addition, the search for specific terms “quercetin”, and “anthocyanins”, “honey polyphenols”, “preprobiotics” and “S-allylcystein” in combination with “prevention/treatment” and/or “Alzheimer’s disease” was carried out to identify the action mechanism of these compounds in AD. Also, we limited the search to studies published after 2000 in English for a comprehensive search strategy.

## 3. Mechanistic Insights of Potential Nutraceuticals in AD

### 3.1. Quercetin/Apple

Quercetin belongs to a subcategory of flavonoids called flavanols and is one of the most consumed molecules of these compounds within the human diet, being consumed on a daily average of 5 to 40 mg [17,18]. The chemical structure of quercetin consists of three ring structures and five hydroxyl groups. It can cross the blood brain barrier, which is an important feature in the neurodegenerative disease’s context [5,19]. Quercetin has multiple properties that are beneficial for human health including anti-inflammatory and antioxidant capacities [19]. The latter is especially important in the context of neurodegenerative diseases because the brain is an organ susceptible to oxidative stress due to its high composition of unsaturated fatty acids, high oxygen consumption, and low antioxidant capacity [19].

Flavonoids are some of the major categories of antioxidants that can be found in apples, being quercetin one of the most important ones within this classification [7]. Quercetin can be extracted from the whole fruit, but apple peel contains greater amounts of this substance rather than the flesh of this fruit [7]. It was estimated that apples contain 2.1 to 7.2 mg/100 g of quercetin, which is mostly found in its glycoside form that is soluble in water [20]. All the above-mentioned effects are summarized in Figure 1.

Some studies showed that quercetin is capable, in low concentrations from 5 to 10 μM, of reducing the damage and cell death caused by treatments with H_2_O_2_ and Aβ in cell models of PC12 cells and primary neuronal cultures, respectively [19]. Moreover, studies in the murine triple-transgenic Alzheimer’s models showed that, treatment with quercetin can significantly reverse pathological processes, such as β-amyloidosis, tauopathies, astrogliosis, and microgliosis. The test animals also improved in their memory and learning performances [19]. It was also shown in in vivo studies with rodents that quercetin’s administration (0.5 to 50 mg/kg) has a protective effect against oxidative stress and against the damage caused by various neurotoxic components [18].

In silico analysis also revealed that quercetin has a superior inhibitory capacity over AChE than that of conventional drugs used to treat AD, because quercetin (especially in its methylated form azaleatin) presents a stronger union with the active site of this enzyme as compared to that of conventional drugs used in clinical practice [19].

The antioxidant properties of quercetin are mostly given by its capacity of scavenge free radicals, its metal chelating ability, and its capacity to protect neurons against the toxicity of metals [17,18]. Quercetin can modulate enzymatic systems, such as the nitric oxide synthase, and transcriptional factors, such as NF-κβ and Nuclear factor erythroid 2-related factor 2 (Nrf-2), that induce genes that code for detoxifying and antioxidant proteins [17,18]. Quercetin can also modulate pathways that are involved in cognition, neurogenesis, and neuronal survival, such as PI3K/Akt, tyrosine kinases, Protein kinase C (PKC), and mitogen-activated protein kinase (MAPK) [17]. The activation of the Nrf2-ARE pathway has a protective effect in neurons against the damage caused by oxidative stress and against cell death; new evidence even suggests that this pathway can modulate the formation and degradation of misfolded protein aggregates present in neurodegenerative diseases such as AD [18].

Quercetin also influences the mitochondria, decreasing the dysfunction in this organelle, reducing Reactive oxygen species (ROS) production and restoring the mitochondrial membrane’s potential and production of ATP [21]. Quercetin also regulates the expression of AMP-activated protein kinase (AMPK), which has a very important role in modulating energy metabolism and reducing ROS production [21]. Another important property of the AMPK in the context of AD, is that these proteins reduce the deposition of Aβ, induce its clearance, and regulate the processing of its precursor protein APP [21].

In this context, quercetin has also an anti-inflammatory effect due to scavenging free radicals and ROS [22]. It was also demonstrated that quercetin can inhibit the expression of TNF-α at the gene expression level by modulating the activity of NF-κβ [6]. In glial cell models induced by Lipopolysaccharide (LPS), quercetin can reduce the mRNA levels of TNF-α y IL-1α, and in neurons and microglia cocultures, quercetin reduces the apoptotic neuronal death induced by microglial activation [22]. Besides, quercetin is involved in promoting autophagy, which is a very important process in maintaining the integrity of the central nervous system, and has a neuroprotective effect [18]. Quercetin can also activate SIRT1 protein which, in turn, can suppress Bax-dependent apoptosis and inhibit proapoptotic transcriptional factors [18].

In the context of AD, in vitro studies using quercetin, probed that this compound has an effective capacity inhibiting protein aggregation of Aβ, tau protein and α-synuclein by stabilizing the oligomeric forms of these misfolded proteins and through this inhibiting fibril growing [5].

Quercetin can also, due to its chemical structure and interaction with factors like BACE-1 and NF-κβ, inhibit the formation of Aβ oligomers and destabilize its fibrils, reducing the neurotoxic effects of this protein aggregates [21]. In another study, using HT22 hippocampal neurons, pretreatments with quercetin demonstrated to inhibit tau hyperphosphorylation [5]. This compound also has the capacity of inhibiting the activity of the CDK5 enzyme, a key component in the regulation of tau [5]. Furthermore, studies involving the triple-transgenic mouse models of Alzheimer’s disease showed that quercetin can reduce the levels of NFTs, Aβ and cognitive impairment in these mice [5].

Another action of quercetin has to do with the process of cellular senescence, which is defined as a permanent arrest of the cell cycle [23]. This process occurs under different conditions such as tissue remodeling in the context of development or after tissue damage. The senescence can also decrease the regenerating ability of the organic tissue and cause inflammation [23]. Cellular senescence is an important process in AD, because it can occur during aging neurons, astrocytes, and microglia that is characterized by the production of inflammatory substances and decreased functionality of tissues/organs [24]. In this context, quercetin was shown in a selective way to eliminate senescent cells in the brain of Alzheimer murine models, suggesting that quercetin has a senolytic activity [24,25].

Within the human diet, fruits and vegetables are important sources of antioxidants and other important substances that are beneficial for health [26]. Apples were proved to be a major source of antioxidants due to its high content of these kind of compounds and due to the high level of consumption of these fruit on the human population [26]. Moreover, several beneficial effects were attributed to this plant, such as anti-inflammatory, antiulcer, and neuroprotective effects [27].

### 3.2. Anthocyanins/Berries

In general, berries are characterized by their high content of minerals, vitamins, dietary fiber, phenolic compounds, and organic acids. However, anthocyanins (ANT) are the main bioactive compounds considered a water-soluble dye. Red wine, blueberry, bilberry, cranberry, elderberry, raspberry, strawberry, maqui, and calafate (endemic Patagonian fruit) are rich sources of natural dietary anthocyanins. Berry extracts were associated with protective effects against AD and other disorders [20,28].

Functional studies in humans associated the intake of berries with slower rates of cognitive decline in elderly subjects, suggesting the protective role of ANT on different cognitive functions [29,30]. The 30 mL blueberry supplementation (387 mg ANT) in healthy older adults showed significant increases in brain activity within areas associated with cognitive function (Brodmann areas, precuneus, anterior cingulate, and insula/thalamus) [31]. A randomized, double-blind, placebo-controlled trial, the older adults with cognitive complaints improved cognition after the long-term supplementation (24-weeks) with blueberry [32]. This shows that ANT-rich berries supplementation has neurocognitive benefit in this at-risk population for dementia [30]. However, the preventive effect will depend on the amount and ANT-structure (aglycone or its glucoside conjugated).

Anthocyanins-berries have interesting pharmacological activities, such as antioxidant and anti-inflammatory, and improve neuronal and cognitive brain performance [29,31]. Regarding the mechanism of action, it was proposed that ANT inhibit tau hyperphosphorylation and activation of GSK-3β induced by Aβ in PC12 cells [5,33]. Structurally, the planar aromatic ring of anthocyanins is essential to inhibit heparin-induced filament formation of tau protein [34]. Other studies showed that inhibition of oxidative stress and neuroinflammation are two critical mechanisms by which ANT produce protective effects in AD prevention or treatment [35] (Figure 2). Long-term, the ANT can upregulate p-PI3K, p-Akt y p-GSK-3β expression, decrease ROS and Malonaldehyde (MDA), and increase Nrf2 nuclear translocation and glutathione cysteine ligase modulatory subunit (GCLM) and HO-1 expression in the hippocampus of APP/PS1 mice [35].

### 3.3. Polyphenols/Honey

Honey was studied since the early 1970s, due to its nutraceutical properties that include antibacterial, bacteriostatic, anti-inflammatory, and wound and sunburn healing activities [36]. In addition to those properties, novel studies demonstrated several antioxidant and nonperoxide-dependent properties [37]. One of the main reasons behind these properties are the polyphenols present in it [38,39], which can also provide highlights regarding the honey’s botanical origins [40].

The latter is important in neurodegenerative diseases such as AD, Parkinson’s disease (PD), Huntington’s disease (HD), and multiple sclerosis (MS) [39]. In all cases, an increased oxidative stress due to the depletion of antioxidants, neuro-inflammation, prions, protein and mitochondrial dysfunction, glutamatergic excitotoxicity, and genetic alterations lead to a dysfunction or death of nerve cells [41]. Accordingly, polyphenols found in honey can prevent neurodegenerative disease in several ways [42]: (i) antioxidant effect in neurons [43]; (ii) enhancement of neuronal function and regeneration [44]; (iii) protection of neurons from Aβ-induced neurotoxicity [45]; (iv) protection of hippocampal cells against nitric oxide-induced neurotoxicity [46]; and (v) modulation of neuronal and glial cell signaling pathways [47].

One flavonoid present in honey is Luteolin. This bioactive compound shows neuroprotective activity against microglia-induced neuronal cell death and enhances the spatial working memory via prevention of microglia associated inflammation in the hippocampus of aged rats [48]. In another study, luteolin enhanced basal synaptic transmission while facilitating the induction of long-term potentiation (LTP) by high-frequency stimulation in the dental gyrus of the rat hippocampus through the activation of cAMP response element-binding protein (CREB); thus, it protects synaptic function and restores memory in neurodegenerative disorders [49]. Consistent with its neuroprotective activity, luteolin also protects against β-amyloid-induced toxicity in rat-cultured cortical neurons [50].

Other flavonoid present in honey is Kaempferol. This molecule exhibits a neuroprotective effect in Parkinson’s disease mice models induced by the neurotoxin 1-methyl-4-phenyl-1,2,3,6-tetrahydropyridine (MPTP). The latter leads to behavioral and biochemical alterations similar to PD, such as behavioral deficits, depletion of dopamine and its metabolites, reduction in SOD and glutathione peroxidase (GSH-PX) activities and elevation of MDA levels in the substantia nigra of mice. When kaempferol was administrated to mice every 24 h for 14 consecutive days, the behavioral and biochemical alterations improved substantially. Neuroprotection was confirmed by the histochemical findings, in which kaempferol prevented the loss of TH-positive neurons induced by MPTP [51].

Another polyphenol present in honey, Ferulic acid, promotes a neuroprotective effect during a middle cerebral artery occlusion as it decreases phospho-PDK1, phospho-Akt and phospho-Bad levels, while preventing the increase in caspase-3 levels [52]. Ferulic acid also displayed a neuroprotective effect against oxidative stress associated apoptosis through inhibition of ICAM-1 mRNA expression and by decreasing the number of microglia/macrophages after cerebral ischemia/reperfusion injury in rats [53]. Also, it was demonstrated its anti-inflammatory and antioxidative properties during a transient-focal ischemia in rats [54].

On the other hand, chlorogenic acid also present in honey exerts a neuroprotective effect against methyl mercury-induced apoptosis in pheochromocytoma-12 (PC12) cell lines. In this study, chlorogenic acid prevents the generation of reactive-oxygen species (ROS), suppressing the decreasing action of glutathione peroxidase (GPx) and Glutathione (GSH) and attenuating apoptosis by the activation of caspase-3 [55] (Figure 3). It also reflects neuroprotective effects in scopolamine-induced learning and memory impairment by inhibiting the activity of acetylcholine esterase and MDA in the hippocampus as well as in the frontal cortex in mice, as demonstrated by Kwon et al. (2010) [56].

### 3.4. Prebiotics and Probiotics

It was recently studied the role of the microbiota in relation to neurodegenerative diseases [12]. Furthermore, the gut–brain axis is now a key component in the development of neurodegenerative disorders [12,57]. Thus, prebiotics and probiotics, which are key for the maintenance of a healthy microbiota, should be considered among the novel compounds that could be employed in AD prevention.

#### 3.4.1. Prebiotics

A Prebiotic is “a non-digestible food ingredient that beneficially affects the host by selectively stimulating the growth and/or activity of one or a limited number of bacteria in the colon, and thus improves host health”, a concept applied for the first time in 1995 [58]. They consist mostly in carbohydrates that are resistant to enzymes and secretions of the gastrointestinal tract [59], such as resistant starch and β-glucan. Once they reach the colon, they are fermented by the microflora present in the gut and promote the growth of commensal strains and inhibit the growth of pathogenic bacteria [60].

One of the most studied prebiotics is fructooligosacharide (FOS). This compound is derived from inulin degradation and is found in many fruits and vegetables and is a substrate for the proliferation of *Lactobacillus* and *Bifidobacterium*, part of the microflora. Regarding the cognitive impairment, FOS supplementation to transgenic AD mice increased Glucagon-like peptide-1 (GLP-1), a protein that readily crosses the blood-brain barrier (BBB) and promotes satiety, pancreatic secretions of insulin and slowing of gastric emptying [61]. Since cerebral GLP-1 increases, it improves central nervous system (CNS) insulin resistance, which consequently decrease neuronal cell death derived from the impaired glucose metabolism observed in AD. FOS supplementation also influences neuroplasticity through the expression of synapsin-1, which coats synaptic vesicles and is also employed as a marker of neuronal activity. In AD patients, synapsin-1 levels considerable decrease. But supplementation with FOS in AD mice lead to restoration of physiologically normal synapsin-1 levels compared to that of the controls [62]. Also, it was demonstrated that *B. longum* accompanied with FOS decreases C- reactive protein (CRP), TNF-α, serum Aspartate transaminase (AST) levels, serum endotoxin, steatosis, HOMA-IR, and the nonalcoholic steatohepatitis activity index significantly [63].

Another prebiotic extensively studied with promising results is xylooligosaccharides (XOS). This compound generates from oligomers of xylan and is the most abundant biopolymer in the plant kingdom [64]. Due to its availability and anti-inflammatory properties, XOS was an excellent candidate to test in cognitive impairment. Indeed, XOS supplements administration to APP/PS1 mice suffering from hepatectomy-induced postoperative cognitive dysfunction (POCD), a common comorbidity of AD, improved its cognition [65]. After a POCD intervention, the most common symptoms observed are memory loss, lack of balance and executive functions due to neuroinflammation and a reduction in BBB integrity. Supplementation with XOS to the operated mice attenuated the microbiota fluctuations, especially in *Bacteroidetes* and *Lactobacillus* genera, to name a few. The latter augmented the richness of the gut microbiota, and consequently, attenuated intestinal inflammation. Indeed, levels of the proinflammatory cytokines IL-1β and IL-6 as well as the immunosuppressive cytokine IL-10 decreased [65].

After surgery, weakening of the BBB and epithelial barrier was observed in AD mice, due to a decrease in tight junction proteins zonulin-1 (ZO-1). Administration of XOS increased ZO-1 in both epithelial and hypothalamic tissue, suggesting a relationship between a “leaky gut” and a more permeable BBB. Indeed, Transmission Eelectron Microscopy (TEM) imagery of the BBB demonstrated that supplemented AD mice showed a comparable BBB composition and structure to control subjects [65].

#### 3.4.2. Probiotics

The current definition of probiotics is “live microorganisms which when administered in adequate amounts confer a health benefit on the host” [66]. Among the most important ones are Lactobacilli (e.g., L. acidophilus, *L. rhamnosus*) and *Bifidobacteria* (e.g., *B. bifidum*, *B. infantis*) which are considered “functional foods” [67]. There are several ways this commensal microorganism can affect the neurodegenerative diseases, and AD in particular.

They affect the CNS biochemistry directly, due to the influence in the levels of several key regulators such as Brain-Derived Neurotrophic Factor (BDNF), G-Aminobutyric Acid (GABA), Dopamine (DA), and serotonin (5 hydroxytryptamine; 5 HT), consequently affecting mind and behavior [68]. This would be explained in part by the gut–brain axis relationship, in which a bidirectional communication occurs between the gut and brain. In that regard, tryptophan and short chain fatty acids produced by probiotic bacteria that indirectly modulate CNS function [61,69]. They also modulate the immune system, diminishing the proinflammatory cytokine production and swelling [70]. Indeed, tryptophan and short chain fatty acids can cooperate with the immune system and regulate cellular immune reactions [61,71,72].

Regarding brain amyloidosis, in a study by Cattaneo et al. (2017) [73], the bacteria analyzed were selected due to their proinflammatory (*Escherichia*/*Shigella* and *Pseudomonas aeruginosa*) or anti-inflammatory (*Eubacterium rectale, Eubacterium hallii, Faecalibacterium prausnitzii, and Bacteroides fragilis*) profile. Consequently, pro- (e.g IL-1β, TNF.α) and anti-inflammatory (IL-4, IL-10, IL-13) cytokine expression was evaluated and corelated to the gut microbiota composition. In this case, it was observed that and increase Escherichia/Shigella and a decrease in *Eubacterium* rectal correlated with changes in the cytokine profiles of the cognitively impaired and amyloid positive patients, leading to a proinflammatory state seen in AD patients [73]. This is consistent with an increase in IL-6, CXCL2, NLRP3, and IL-1β (proinflammatory profile), as well as a decrease in IL-10 (anti-inflammatory cytokine). All the latter sustains the hypothesis of the inflammation modulation of the gut microbiota.

Some probiotic species from *Lactobacillus* and *Bifidum* have demonstrated the ability of Ferulic Acid in large quantities [12]. Furthermore, in preclinical studies, pretreatment with FA was shown to reverse neuroinflammation in transgenic AD-mice, as well as decrease hippocampal and cortical levels of Aβ fibrils when compared to controls lacking the probiotic produced phenolic compound [74].

One interesting probiotic action is through a novel fermentation technology. Kefir is a dairy product similar to liquid yogurt, is obtained through the fermentation due to the action of a group of yeasts (fungi) and bacteria (*lactobacilli*) [75]. Recently, the beneficial effects of kefir in AD were assessed: it was demonstrated that kefir ameliorates cognitive impairment in streptozotocin induced mice model of AD [76]. The latter would be explained, at least in part, by the gut microbiome modulation and enhancement of the insulin/PI3K/Akt pathway, which was demonstrated in another model of sporadic AD [77]. All the latter illustrates the importance of the gut–brain axis on the onset and progression of AD. Some of the most important effects regarding pre/probiotics are summarized in Figure 4.

### 3.5. SAC/Garlic

Garlic (*Allium sativum*) and extracts prepared from fresh garlic are commonly used for medical purposes; moreover, fresh garlic was used as a source of herbal medicine for many years [78]. Among many beneficial effects that this plant has for the human health, compounds found in fresh and garlic extracts were proved to have neuroprotective, antioxidant, and synapto-preservative properties that are important in the context of neurodegenerative diseases such as AD [79].

Fresh garlic can cause some negative effects when it is consumed raw, such us anemia, disruption of the gut microbiota, serum protein levels alterations, and indigestion. [78] That is why nowadays garlic extracts such as aged garlic extract (AGE) and black garlic are used for medicinal purposes, because the aging processes carried out to produce them overcome this side effects and stabilize most of the beneficial constituents of fresh garlic [78]. One of the mayor compounds present in these aged extracts of garlic and black garlic is S-allylcystein (SAC), a compound derived from allicin that is present in fresh garlic [78,80].

Some studies in Alzheimer murine models treated with AGE and SAC demonstrated an important capacity of these compounds that decrease Aβ loads and toxicity in addition to antioxidant properties and the capacity of ameliorating tau pathologies [79]. Among the anti-Aβ activity of these substances, some in vitro studies showed that treatments with SAC can disaggregate these proteins, and in vivo studies in transgenic mice treated independently with AGE and SAC resulted in a decrease in amyloid plaques in the mice brains [79]. Homogenates of these transgenic mice brains even showed a reduction in the intracellular precursor of Aβ (APP) when treated with AGE [79]. SAC can also prevent the neurodegeneration caused by Aβ toxicity in the hippocampus by ameliorating endoplasmic reticulum stress and by inhibiting the activation of caspase three, which decreases synaptic function and postsynaptic density [79]. AGE treatments were even capable to improve learning and memory deficits measured by cognitive tests in transgenic mice due to Aβ toxicity [78].

Treatments using AGE and SAC separately, showed in Alzheimer mice models that these substances are capable of decreasing tau phosphorylation mostly by inhibiting the activity of GSK-3β [79]. These garlic derived compounds also have a role in diminishing neuroinflammation because of their capability of scavenging ROS and due to their anti-Aβ activity, that results as well in a decline of tau hyperphosphorylation [79]. This ROS-scavenging activity is proposed to happen due to the modulating ability of garlic derived compounds of modulating intracellular levels of GSH, which in turn has a very important role in the cellular protection against oxidative stress generated by ROS [79]. Studies using cellular cultures also showed that SAC treatments can inhibit NF-κβ activation, which is an important mediator in Alzheimer’s disease inflammation due to its role in producing inflammatory cytochemokines [79]. Other studies show that AGE treatments can even decrease IL-1β levels in the hippocampus [78] (Figure 5).

In studies using LPS-treated cell cultures, aged garlic extracts were proved to have the ability to reduce levels of NO and inflammatory cytokines, such as IL-6 and TNF-α, in addition to decreasing the expression of COX-2 and iNOS [80].

### 3.6. Palmitoylethanolamide (PEA)

In the context of nutritional supplements, the actions of Palmitoylethanolamide (PEA) are notable. This is an endocannabinoid-like lipid mediator with extensively documented anti-inflammatory, analgesic, antimicrobial, immunomodulatory, and neuroprotective effects. It is recommended the use of PEA, as with other nutraceuticals with known mechanisms that prevent neurodegenerative disorders and AD, especially in subjects at risk due to age or to epigenetic antecedents [81]. 

### 3.7. Bowsellic Acid (AKBA)

Another interesting compound is Boswellic Acid, which is a pentacyclic terpenoid obtained from Indian herbal medicine that is produced by plants in the genus *Boswellia* [82]. In particular, 3-Acetyl-11-Keto-Beta-Boswellic Acid (AKBA) was demonstrated to have neuroprotective effects against LPS-induced neuroinflammation through the modulation of miRNA 155 [83]. In agreement with this anti-inflammatory effect, it was demonstrated that coadministration of celeoxib and 3-Acetyl-11-Keto-Beta-Boswellic Acid potentiates the protection against LPS-induced cognitive impairment in mice. Furthermore, another study demonstrated that pre-treatment with Acetyl-11-Keto-Beta-Boswellic Acid have an effect on brain cytokines that finally lead to a decrease in proinflammatory cytokines such as TNF-a and an improvement in cognitive performance in LPS-induced memory impairment in rats [84]. Finally, in AD, it was demonstrated that AKBA possess potent anti-inflammatory and neuroprotective effect. This is potentiated by the inhibition of the acetylcholinesterase (AChE), which improves the level of acetylcholine [85].

## 4. Current Monotarget versus Multitarget Therapies

Currently, none of the pharmacologic treatments available for AD slow or stop the damage and destruction of neurons that cause symptoms and make the disease fatal [86]. According to Food and Drug Administration (FDA), today there are six approved drugs to control AD (memory or cognition), and most of them focus on a single molecular target. Three of these drugs are donepezil, glutamine, and rivastigmine, all of which are cholinesterase inhibitors [86]. The fourth drug is memantine, which exerts its effect asN-methyl-D-aspartate receptor (NMDAR) antagonist and improves temporarily cognitive symptoms by increasing the neurotransmitters [86,87]. The fifth drug is a combination of donepezil and memantine. The sixth was recently authorized: aducanumab, a monoclonal antibody that recognizes the structural conformation of Aβ species and provokes the diminishing amyloid plaques or clearance of the aggregates, but also leads to microglial activation and phagocytosis [87]. Moreover, the effectiveness of these drugs to ameliorate the cognitive performance varies from person to person and is limited in duration [86]. These drugs correspond to the monotargeted category of therapeutic agents. On the other hand, nutraceuticals containing several bioactive molecules are involved in multitarget therapy (Figure 6), which is of interest considering that AD is a multifactorial disease.

Also, there are behavioral and psychiatric symptoms (insomnia, depression) that may develop the patients with moderate and severe stages of Alzheimer’s, but no drug was approved by FDA to treat these symptoms [86]. Although, the physicians may prescribe antipsychotics to treat hallucinations, aggression, and agitation in their patients [88]. However, these were associated with an increased risk of stroke and death in AD individuals, so that their use must be cautious as they are not specific to treating AD symptoms [86]. In this context, there are many therapeutic options to treat and prevention AD among them are the nutraceuticals that act synergistically in several processes in the brain that cause AD and reducing neuropathological damage.

In the context of the multitarget therapeutic approaches [5], nutraceuticals play a major role. Nutraceuticals are defined as a food product or its secondary metabolites that could deliver health benefits (to prevent or treat diseases) in the clinical setting [89]. Nutraceuticals refer to the combination of pharmaceutical and nutrition, which is characterized by multitarget action. Several studies indicated the preventive effect of nutraceuticals against AD (complementary medicine) via regulating neuro-oxidative stress, neuroinflammation, and protein aggregation, enhancing neurogenesis, and regulating mitochondrial function (suppress abnormal mitochondrial dynamics) through various signaling pathways [5,89]. Antitau, antiamyloid, and anti-inflammatory molecules continue to b in focus where presymptomatic interventions are necessary [88]. Among the latter, we can mention:

Brain-Up10^®^ is a formulation of Andean *shilajit* and Vit-B complex—its multitarget action was evaluated in several clinical trials that demonstrated their antioxidant and anti-inflammatory properties as well as its effect in disassemble of tau oligomers [15]. Interestingly this formula has mechanisms to ameliorate cognitive decline, thereby improving apathy and neuropsychiatric symptoms [15].

We believe that treatments with nutraceuticals could help preserve brain function when they are administered early in the disease continuum, either at the MCI due to Alzheimer’s or preclinical Alzheimer’s phase. In the future, a large-scale, well-controlled clinical trial should be conducted with prominent nutraceuticals applied to patients with AD, and it will also be needed to confirm the bioavailability and nutrigenomic data to choose the better anti-AD agent, effective in a clinical setting to treat the complex etiology of AD (see Figure 6).

## 5. Conclusions

We thoroughly revised the principal groups of nutraceuticals that have proven actions for the prevention and treatment of Alzheimer’s disease (AD). This work illustrates the enormous growth of bioactive compounds in the prevention and therapeutic strategies for this disease. Moreover, we also detailed the main evidence supporting a novel multitarget approach regarding Alzheimer’s disease. Current monotarget approaches are evidently not sufficient for the proper treatment of AD patients. In that regard, nutraceuticals are currently a valuable option as they can influence more than one target related to the development/onset of AD. They can modulate several factors related to AD, including modulation of the gut microbiome and, consequently, central nervous system (CNS) biochemistry, antiaggregation properties of Aβ and tau, and anti-inflammatory properties.

## Figures and Tables

**Figure 1 biomolecules-12-00249-f001:**
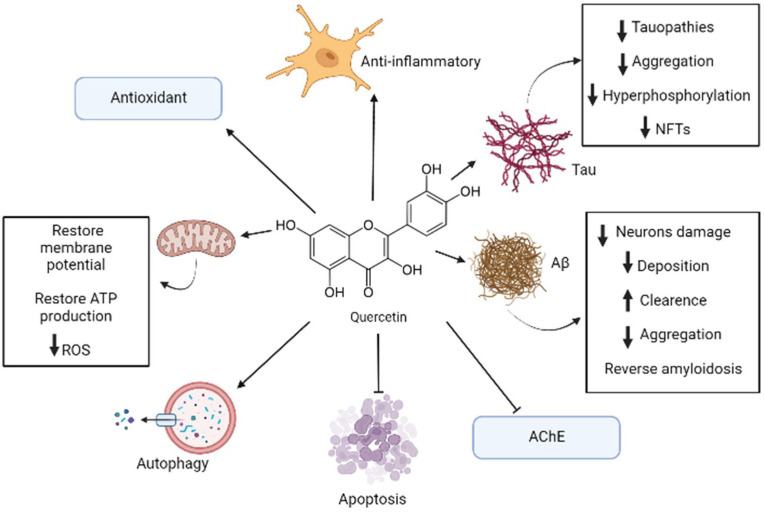
Representative model of anti-Alzheimer’s disease (AD) mechanisms of quercetin. In AD, quercetin induces multitargeted molecular mechanisms including antioxidant, antiapoptotic, anti-inflammatory, antitau, and anti-Aβ action. It is also involved in autophagy promotion, acetylcholinesterase (AChE) inhibition, and reversing mitochondrial disruption. The black arrow up (increase), the black arrow down (decrease). Thin arrow (stimulation) and line (Inhibition).

**Figure 2 biomolecules-12-00249-f002:**
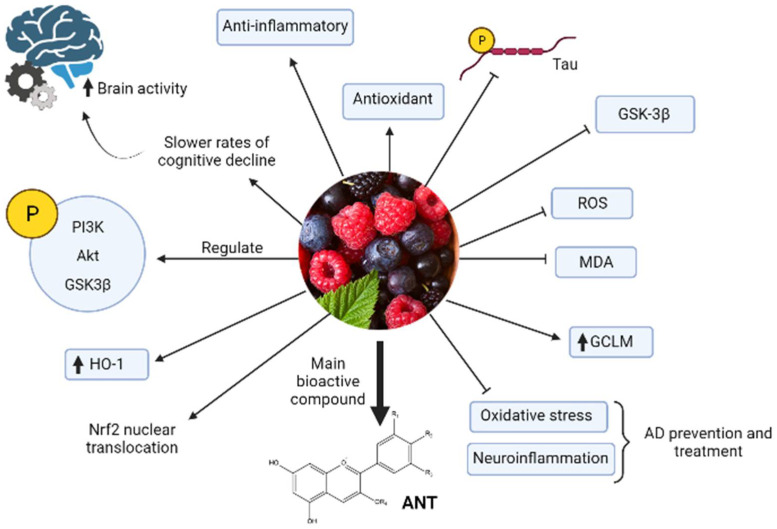
Model of anthocyanin properties against AD. Berries and their main bioactive compounds anthocyanins (ANT) have anti-inflammatory, antioxidant, and neuroprotective properties that make possible prevention and treatment of AD through different mechanisms. The black arrow up (increase), the black arrow down (decrease). Thin arrow (stimulation) and line (Inhibition).

**Figure 3 biomolecules-12-00249-f003:**
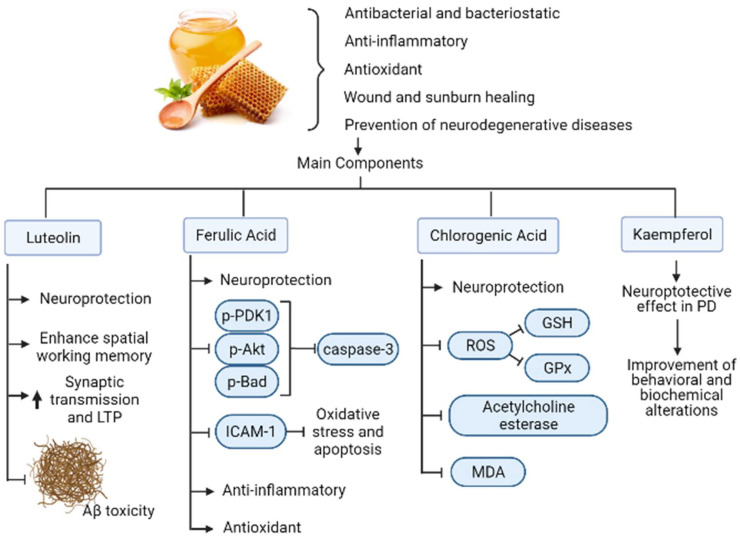
Mechanisms of action of honey polyphenols in AD. Honey has beneficial properties for human health that include antibacterial, anti-inflammatory, antioxidant, wound healing, and prevention of neurodegenerative diseases such as AD. Their main components consist of flavonoids and polyphenols that exert those properties as well as neuroprotective, antiapoptotic, anti-Aβ and synaptic transmission, and memory enhancement activities. The black arrow up (increase), the black arrow down (decrease). Thin arrow (stimulation) and line (Inhibition).

**Figure 4 biomolecules-12-00249-f004:**
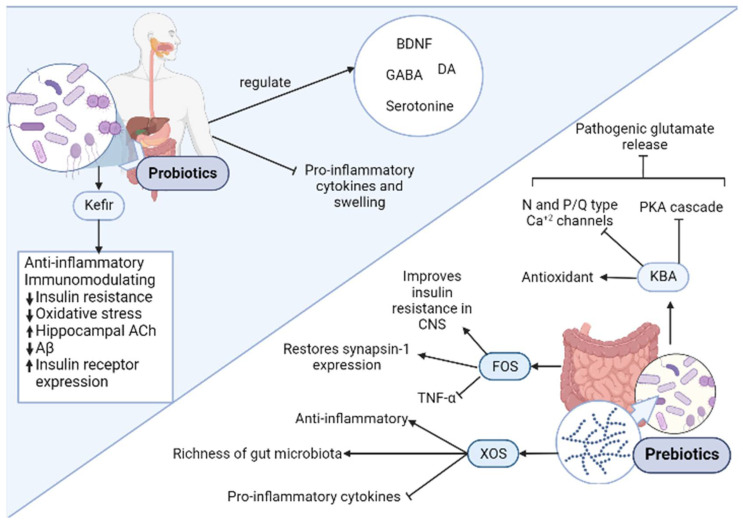
Relevant mechanisms of prebiotics and probiotics in AD. Prebiotics consist of nondigestible components of food that are beneficial to microbiota. These compounds have neuroprotective and anti-inflammatory effects in addition to improving insulin resistance in central nervous system (CNS). Some examples of probiotics are Fructooligosaccharide (FOS), Xylooligosaccharide (XOS), and 11-keto-β-boswellic acid (KBA). Probiotics are live microorganisms that, when administered in appropriate concentrations, benefit the host´s health. These compounds have anti-inflammatory, antioxidant, anti-Aβ, and immunomodulating and other effects related to insulin regulation. They also exert an important role in regulating the levels of relevant components of brain biochemistry. A representative example of a probiotic is kefir, which exerts multiple beneficial health activities. The black arrow up (increase), the black arrow down (decrease). Thin arrow (stimulation) and line (Inhibition).

**Figure 5 biomolecules-12-00249-f005:**
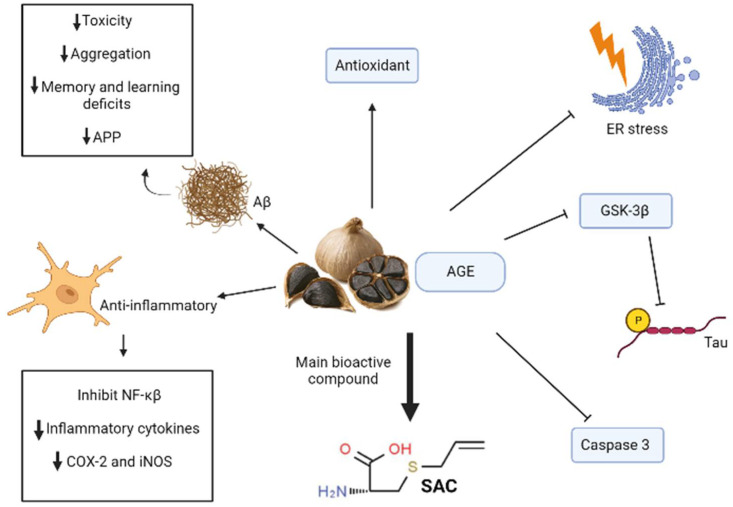
Mechanism of garlic and S-allylcystein (SAC) action in AD. Aged garlic and its bioactive compound S-allylcystein (SAC) elicit anti-inflammatory, antioxidant, anti-Aβ responses and antitau aggregation. Specifically, inhibition of tau aggregation occurs by a reduction in glycogen synthase kinase 3 beta’s (GSK-3β) activity. They also have capacity to inhibit caspase 3 and avoid endoplasmic reticulum stress (ER stress). The black arrow up (increase), the black arrow down (decrease). Thin arrow (stimulation) and line (Inhibition).

**Figure 6 biomolecules-12-00249-f006:**
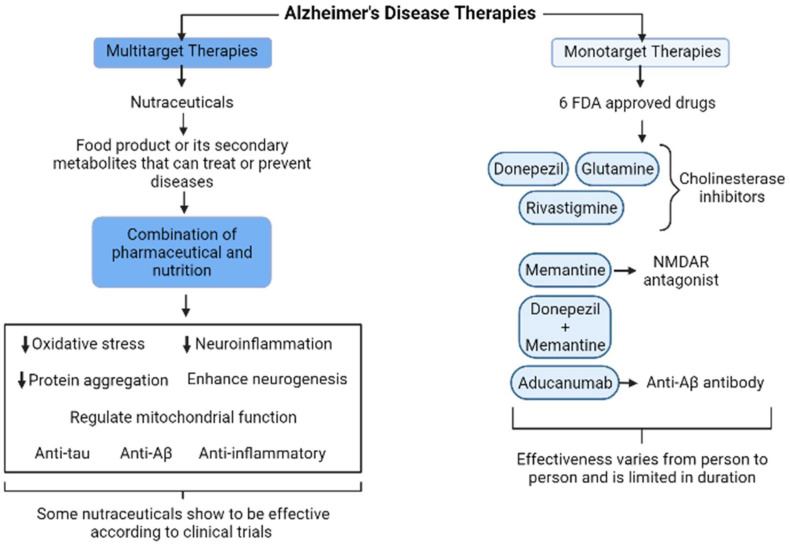
Comparison of monotarget and multitarget therapies for AD. Monotarget therapies consist of drugs that treat just one aspect of AD and tend to vary in effectiveness between patients, and they also have a limited duration. Multitarget therapies consist of nutraceuticals that combine nutrition and pharmaceutical effects, and they tend to act synergistically in processes associated with AD in brain. The black arrow up (increase), the black arrow down (decrease). Thin arrow (stimulation) and line (Inhibition).

## Data Availability

Not applicable.

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
