# Peer review of "Novel Nutraceutical Compounds in Alzheimer Prevention"

_biomolecules, 2022, doi:10.3390/biom12020249_

Round 1

Reviewer 1 Report

Dear Authors,

I have read the manuscript and I send you my comments:

1) Methods are missing, please add the method used in the selection of papers

2) Line 84, please add as reference this manuscript: doi: 10.3390/molecules2501006

3) in the mchanism of ation of quercetin please add and discuss this paper: Int Wound J. 2020 Apr;17(2):485-490. doi: 10.1111/iwj.13299. Epub 2019 Dec 25.

4) Please add the role of Palmitoylethanolamide as neurological treatment  

Author Response

  • Methods are missing, please add the method used in the selection of papers.

Answer: In agreement with review in this revised version we have included statement of methods. “A systematic search for relevant bioactive compounds or nutraceuticals linked to prevention of Alzheimer disease was performed according to the guidelines and items required for Systematic Reviews. The following electronic databases were used to identify pertinent publications: Web of Science, PubMed®, Springer and Google Scholar. The literature search was conducted in September-November 2021. Combinations of the search terms “bioactive compounds”, “nutraceuticals” and/or “Alzheimer’s disease” were used. In addition, the search specific terms “quercetin”, and “anthocyanins”, “honey polyphenols”, “pre-probiotics” and “S-allylcystein” in combination with “prevention/treatment” and/or “Alzheimer’s disease” was used to identify the action mechanism of these compounds in the Alzheimer. Also, we limited the search to studies published after 2000 in English for a comprehensive search strategy” (lines 105-115)

  • Line 84, please add as reference this manuscript: doi: 10.3390/molecules2501006

Answer: We have decided to delete reference as it was considered no relevant for the manuscript.

  • In the mechanism of action of quercetin please add and discuss this paper: Int Wound J. 2020 Apr;17(2):485-490. doi: 10.1111/iwj.13299. Epub 2019 Dec 25.

Answer: The mechanism of quercetin has been analyzed by Int Wound J. 2020 Apr;17(2):485-490. doi: 10.1111/iwj.13299. Epub 2019 Dec 25. However, we decided not to include since the mechanistic approaches, even though their interest, are far beyond Alzheimer´s disease.

  • Please add the role of Palmitoylethanolamide as neurological treatment  

Answer: We thank the reviewer for this important point. In the context of nutritional supplements, it is worth mentioning the actions of Palmitoylethanolamide (PEA). This is an endocannabinoid-like lipid mediator with extensively documented anti-inflammatory, analgesic, antimicrobial, immunomodulatory and neuroprotective effects. It is recommended, like other nutraceuticals with knowns, mechanisms to be used in prevention of neurodegenerative disorders and AD, especially in subjects under risk due to age or to epigenetic antecedents (Clayton et al., 2021). So, as recommended by the reviewer, a paragraph regarding PEA was added to the manuscript (lines 468-476)

Reviewer 2 Report

The review article entitled:NOVEL NUTRACEUTICAL COMPOUNDS IN ALZHEIMER PREVENTION is an interesting article focused on the natural compounds used for management of alzheimer's disease. It could be considered for publication after performing the following comments:

  • More figures and diagrammatic sketches  should be added to show the mechanistic pathways of different group of compounds in alzheimer's disease
  • At least two tables summarizing the collected data showing the different selected compounds and brief description of the mechanism of action
  • A paragraph describing the relation of neuroinflammation and development of alzheimer's disease should be added

More information should be added as the info provided are focused is few, many other important compounds were found to be beneficial in the management of AD, the hereunder compounds and probiotics are found to be beneficial in the management.                                             

Probiotics Fermentation Technology, a Novel Kefir Product, Ameliorates Cognitive Impairment in Streptozotocin-Induced Sporadic Alzheimer’s Disease in Mice

Enhancement of Insulin/PI3K/Akt Signaling Pathway and Modulation of Gut Microbiome by Probiotics Fermentation Technology, a Kefir Grain Product, in Sporadic Alzheimer’s Disease Model in Mice

Role of 3-Acetyl-11-Keto-Beta-Boswellic Acid in Counteracting LPS-Induced Neuroinflammation via Modulation of miRNA-155

Co-administration of 3-Acetyl-11-Keto-Beta-Boswellic Acid Potentiates the Protective Effect of Celecoxib in Lipopolysaccharide-Induced Cognitive Impairment in Mice: Possible Implication of Anti-inflammatory and Antiglutamatergic Pathways

Author Response

  • More figures and diagrammatic sketches should be added to show the mechanistic pathways of different group of compounds in Alzheimer’s disease.

Answer: As suggested by the reviewer, we have included 5 figures with the action mechanism of each nutraceutical compound studied. In addition, we added as Figure 6, a representative sketch that summarizes the mechanistic pathways of mono and multitarget therapies in AD.

  • At least two tables summarizing the collected data showing the different selected compounds and brief description of the mechanism of action

Answer: A detailed description of several nutraceuticals’ compounds with antiaggregating, antioxidant, anti-inflammatory and other effects linked to Alzheimer was already summarized in Calfio et al.2020. Instead of Tables, we have decided to include representative figures. We included the Figures 1,2,3, 4, 5 and 6 with the action mechanism of major nutraceutical compounds reviewed.

  • A paragraph describing the relation of neuroinflammation, and development of Alzheimer’s disease should be added.

Answer: As suggested by the reviewer, the following paragraph was added to the manuscript “Neuroinflammation is one of the major causes for Alzheimer’s disease. The mechanisms on how the inflammatory process occurs in the human brain starts with the so named “damage signals”, which interferes with the cross-talks neuron-glia. As a consequence of that, activated microglia produce NFkB leading the synthesis of proinflammatory mediators that finally signal on neuronal receptors, with reactivation of proteins kinases responsible for tau hyperphosphorylation. In a search of nutraceutical bioactive principles, we can find compounds with tau antiaggregant activity, as well as compounds with antioxidative and anti-inflammatory activities” (lines 56-63)

  • “ More information should be added as the info provided are focused is few, many other important compounds were found to be beneficial in the management of AD, the hereunder compounds and probiotics are found to be beneficial in the management. Kefir Product, Keto-Beta-Boswellic Acid                  

Probiotics Fermentation Technology, a Novel Kefir Product, Ameliorates Cognitive Impairment in Streptozotocin-Induced Sporadic Alzheimer’s Disease in Mice.

Enhancement of Insulin/PI3K/Akt Signaling Pathway and Modulation of Gut Microbiome by Probiotics Fermentation Technology, a Kefir Grain Product, in Sporadic Alzheimer’s Disease Model in Mice.

Role of 3-Acetyl-11-Keto-Beta-Boswellic Acid in Counteracting LPS-Induced Neuroinflammation via Modulation of miRNA-155.

Co-administration of 3-Acetyl-11-Keto-Beta-Boswellic Acid Potentiates the Protective Effect of Celecoxib in Lipopolysaccharide-Induced Cognitive Impairment in Mice: Possible Implication of Anti-inflammatory and Antiglutamatergic Pathways”.

Answer: As suggested by the reviewer, we added the properties of Kefir in the probiotic section and 3-Acetyl-11-Keto-Beta-Boswellic Acid (AKBA) in the prebiotic section. In the case of AKBA, is a potential nutraceutical that have demonstrated to have neuroprotective effects against LPS-induced neuroinflammation with interesting properties to Alzheimer’ disease. These nutraceuticals also were included in the Figure 5.

Round 2

Reviewer 1 Report

None

Reviewer 2 Report

No additional comments